# Welfare and Clinical Assessment on Physical Captures Followed by Anesthesia in Apennine Chamois (*Rupicapra pyrenaica ornata*)

**DOI:** 10.3390/ani13030460

**Published:** 2023-01-28

**Authors:** Simone Angelucci, Antonio Antonucci, Fabrizia Di Tana, Marco Innocenti, Giovanna Di Domenico, Luca Madonna, Camilla Smoglica, Cristina Esmeralda Di Francesco, Jorge Ramón López-Olvera

**Affiliations:** 1Wildlife Research Center, Maiella National Park, 65023 Caramanico Terme, Italy; 2Faculty of Veterinary Medicine, University of Teramo, 64100 Teramo, Italy; 3Wildlife Ecology and Health Group and Departament de Medicina i Cirurgia Animals, Universitat Autònoma de Barcelona (UAB), 08193 Barcelona, Spain

**Keywords:** acepromazine, blood gas analysis, capture, chemical immobilization, conservation, medetomidine–ketamine, physical capture, *Rupicapra pyrenaica ornata*, stress

## Abstract

**Simple Summary:**

Apennine chamois (*Rupicapra pyrenaica ornata*) is a vulnerable subspecies, and creation of new colonies is required for its conservation. Capturing mountain ungulates is not only logistically challenging, but also requires careful planning and clinical monitoring in order to ensure animal welfare and survival. This study reports collective physical capture followed by anesthesia of 21 Apennine chamois for translocation with conservation purposes. The chamois were captured with up-net^®^ (Fondazione Universitaria “Centro Ricerche sulla Gestione della fauna selvatica”—CE.RI.GE.FA.S., Sampeyre—CN) or box-traps, then anesthetized with medetomidine–ketamine–acepromazine and monitored with advanced field techniques, including venous blood gas analysis, which was used for the first time on mountain ungulates in Europe. Despite the risk of metabolic acidosis and catching myopathy, 19 chamois survived and adapted to the new environment to found new colonies, which were monitored through GPS radiocollar data of all released chamois.

**Abstract:**

The Apennine chamois (*Rupicapra pyrenaica ornata*) is one of the rarest subspecies in Italy, listed in Annexes II and IV of the Habitats Directive and currently included as a vulnerable subspecies in the International Union for Conservation of Nature (IUCN) Red List. The Maiella National Park population has recently been defined as a source population for reintroduction into other parks. Since collective captures allow for better selection of target animals for the establishment of new colonies, the aim of this study is to evaluate the physiological conditions and animal welfare in free-ranging Apennine chamois after collective physical capture followed by chemical immobilization with medetomidine 0.054 mg ± 0.007, ketamine 2.14 mg ± 0.28, and acepromazine 0.043 mg ± 0.006. Twenty-one Apennine chamois (18 females and 3 males) were captured and translocated for conservation purposes. The effects of capture and anesthesia were evaluated using clinical variables, hematology, serum biochemistry, and venous blood gas analysis, the latter being used in the field for the first time in chamois capture. A risk of metabolic acidosis and capture myopathy was identified, although it did not compromise the survival of 19 chamois, which adapted to novel environments and founded new colonies, as verified through GPS radiocollars. The protocol applied in this study represents an innovative approach to assessing animal physiology and welfare in collective mountain ungulate captures, useful for improving management activities for conservation purposes.

## 1. Introduction

The Apennine chamois (*Rupicapra pyrenaica ornata*) is a chamois subspecies endemic to the Central Apennines. It is considered a priority subspecies in the Council Directive 92/43/EEC on the conservation of natural habitats and of wild fauna and flora, and it is listed in the Appendix II of the Convention on International Trade in Endangered Species of Wild Fauna and Flora [1]. In 1996, it was classified as “Endangered” in the IUCN red list, but the reintroduction actions carried out in Central Italy have achieved a classification change to “Vulnerable” [2]. In agreement with the aims of the LIFE Project Coornata [3], the chamois population in Maiella National Park (MNP) was selected as a source for reintroduction activities in the Monti Sibillini National Park and in the Sirente Velino Natural Park. 

Any capture or handling operation of wildlife endangers animal welfare and survival, both because of the risk of physical injuries and by eliciting a stress response that can lead to a potentially fatal syndrome known as capture myopathy [4,5,6,7,8,9,10,11,12]. This capture-associated distress is linked to hematological and blood biochemical short- or long-term side effects that are directly related to the capture method applied [13,14,15]. Blood variables, together with other physiological values, are sensitive indicators of alterations in animal homeostasis during capture and stress episodes in wild ungulates [16,17]. They also allow for the evaluation of the physiological, ecological, and nutritional status of wild populations [18,19,20]. This is even more relevant in the case of capture, handling, and translocation operations for conservation purposes in protected vulnerable non-hunted populations, as is the case for the Apennine chamois, since capture consequences may affect the outcome of the individuals during capture and after release, thus compromising the success of the entire management and conservation operation. Additionally, such handlings also provide a unique opportunity to assess the health status of the population.

As for physical capture methods, chamois have been captured with snares, box-traps, enclosures, and diverse net types, including drive-nets and up-net^®^ [19,21,22,23,24]. Chemical immobilization of chamois (*Rupicapra* spp.) has been performed using xylazine alone [25,26,27,28]; xylazine and ketamine [29,30,31,32]; xylazine and tiletamine–zolazepam [33]; xylazine and carfentanyl antagonized with naloxone and yohimbine [34]; medetomidine and ketamine antagonized with atipamezole [35,36,37]; and etorphine and acepromazine antagonized with diprenorphine [38]. Additionally, acepromazine has been used alone as a sedative in chamois captures [22,39,40]. Stress related to physical capture and restraint in wildlife has been reported to increase heart rate, body temperature, red blood cell count, mean concentration of hemoglobin, hematocrit, serum concentrations of glucose, lactate, creatinine, urea, and bilirubin, and to affect the electrolyte balance, increasing serum potassium [17,22,39,40,41,42]. Stress also induces the so-called biphasic stress leukogram, consisting of initial neutrophilia and lymphocytosis induced by catecholamines and followed by neutrophilia, lymphocytopenia, eosinopenia, and monocytosis during the cortisol-predominant resistance phase [17,22,39,40,41,42,43,44].

Finally, the serum activities of alanine aminotransferase (ALT), aspartate aminotransferase (AST), creatine kinase (CK), and lactate dehydrogenase (LDH) constantly increase during capture and mechanical containment of the animal [17,22,39,40,41,42]. 

Few data evaluating the effects of different capture methods on hematological and biochemical parameters are available for Pyrenean chamois (*Rupicapra pyrenaica pyrenaica*) [17,22,39,40], and none for Apennine chamois. The aim of this study is to assess animal welfare by physical collective capture of Apennine chamois, followed by anesthesia, as is required for translocating chamois groups to found new colonies, using clinical variables, hematology, serum biochemistry, and venous blood gas analysis.

## 2. Materials and Methods

### 2.1. Study Area

The Maiella National Park is a protected area of 74,095 hectares in central Apennines and includes 39 municipalities in the provinces of L’Aquila, Chieti, and Pescara in the Abruzzo Region (Italy). In the Maiella massif, Apennine chamois reintroduction started in 1992, and since then, the population has shown exponential growth that is typical of a colonizing population, reaching a minimum number of 839 individuals counted in 2013 [33,45]. 

All the chamois capture sessions were carried out in good weather conditions during the summer season, from August 2012 to August 2014, in selected sites (Table 1). Since the capture methods used were operator-activated and therefore selective [5,21,24], five criteria were used to select suitable herds before triggering the traps: stability, herd size, presence of target animals (mostly young and adult females), use of known areas, and feasibility of monitoring activity [46].

### 2.2. Capture Methods

Eight capture sessions were carried out using physical capture methods followed by chemical immobilization [24,47,48]. Five capture operations were carried out with up-net^®^ [24], and three capture operations with box traps [49,50].

The up-net^®^ trap is characterized by an electro-mechanical system that raises the nets from the ground to form an enclosure [24], while the box-traps used were mobile wooden structures with rigid net walls. These had open entrances that closed when activated remotely, with descending nets where the chamois become entangled when trying to exit the trap. Salt blocks were used as baits inside both trap types [5,21,49]. The box-traps, in some cases, were modified and adapted to be used in caves and rocky ravines usually visited by the animals, as previously reported [49,50]. Once entangled in the electro-mechanical trap or the box traps, all the chamois were chemically immobilized using 50 µg/kg of medetomidine (Domitor^®^, Vetoquinol, Italy) combined with 2 mg/kg of ketamine (Imalgene^®^, Merial, Italy) and 0.04 mg/kg of acepromazine (Prequillan^®^, Fatro, Italy), based on the estimated weight of the captured animals [49]. The anesthetic mixture was administered by hand via intramuscular injection into the shoulder muscles, using 5 mL syringes with 1.1 × 25 mm hypodermic needles. The time elapsed between the activation of the trap and the administration of the drug, immediately after the arrival of the veterinarian, was registered for each captured chamois. The induction time (T0) was recorded as the time interval between the hand administration of the anesthetic mixture and the absence of voluntary movements of the chamois.

### 2.3. Clinical Monitoring and Management

Once immobilized, the chamois were placed in right lateral recumbency, blindfolded, and marked with plastic ear tags. The chamois were clinically examined for disease signs and lesions, then surveilled using a multiparametric monitor and blood gas analysis. A monitoring protocol was used to record the clinical findings at specific time intervals after T0, as follows: T1 (15–25 min), T2 (34–37 min), and T3 (45–54 min). The heart rate (HR), respiratory rate (RR), and rectal temperature (RT) were determined using a stethoscope and a digital thermometer. Atipamezole (Antisedan^®^, Vetoquinol, Italy) at 0.12–0.16 mg/kg was intramuscularly injected to reverse sedation at time T4. The awakening time (T5) was defined as the time between the injection of the antagonist and the recovery of the quadrupedal station. Blood samples were collected from the jugular vein between 15 and 45 min after T0, with the use of disposable 5 mL syringes and 0.8 × 25 mm needles. Two mL of each sample were placed in commercial tubes with ethylene diamine tetraacetic acid K3 (EDTA) as an anticoagulant for hematologic analyses, and the remaining blood was placed in serum collection tubes and allowed to clot at room temperature. Once clotted, the sample was centrifuged at 3000 G for 10 min and the serum was frozen at −20 °C until the hematological and biochemical analyses were performed.

The chamois were fitted with GPS/GSM radiocollars (Followit^®^, Lindesberg, Sweden, or Vectronic^®^, Berlin, Germany), programmed with an intensive monitoring protocol that would allow us to verify the animals’ health status and survival rate in the post-capture period. The chamois deceased either during the capture sessions or after release were recognized through a mortality signal from the GPS radiocollar after four hours of immobility. These chamois were stored refrigerated (2–8 °C) until necropsies were carried out at the local Istituto Zooprofilattico (National Public Health Service) within 24 h of death.

All the captures and procedures performed on the chamois were part of the required management of the species for conservation purposes, and no capture, management, nor procedure was performed on the chamois specifically to conduct this study.

### 2.4. Hematological and Biochemical Analyses

Hematological analyses were carried out by the IZSAM—Istituto Zooprofilattico Sperimentale dell’Abruzzo e del Molise “G. Caporale” (emocromo citometrico, cytochemical method, SOP 003-rev 0-07) within 24 h of collection. The hematologic variables which were measured included RBC (red blood cell count); hemoglobin concentration (HGB); hematocrit (HCT); mean corpuscular volume (MCV); mean corpuscular hemoglobin (MCH); mean corpuscular hemoglobin concentration (MCHC); white blood cell count (WBC); lymphocyte (LYM) count and proportion; monocyte (MID) count and proportion; and granulocyte (GRAN) count and proportion. The blood biochemistry variables which were measured included the serum concentrations of glucose, cholesterol, triglycerides, urea, creatinine, total bilirubin, total protein, albumin, and globulins; the serum activities of creatin kinase (CK), alanine aminotransferase (ALT), aspartate aminotransferase (AST), alkaline phosphatase (ALP), gamma-glutamyltransferase (GGT), and amylase; and the serum concentrations of calcium, phosphorus, sodium, and potassium (K+). The biochemical analyses were carried out at the Maiella National Park laboratory—Wildlife Research Center with the Vetscan^®^ VS2 2.1.5, except for cholesterol (enzymatic-colorimetric, e-c), triglycerides (kinetic-colorimetric, k-c), AST (Aspartate Aminotransferase—kinetic), and GGT (γ-Glutamyltransferase—kinetic, k). The hematologic variables were determined with a flow cytometer. The CK activity was determined in nine chamois by the University of Milan within the Life Coornata Project [3,46]. Blood venous gas analysis in the field was performed with handheld blood gas analyzer i-STAT 1^®^ (Abbott Laboratories, Abbott Park, IL, USA) using CG4+ and EC8+ cartridges immediately after blood collection. The logistic constraints on field sampling in mountain environments prevented the obtention of a sufficient volume of blood to determine all the variables analyzed in all the captured chamois.

### 2.5. Statistical Analysis

The analysis of variance (ANOVA) was used to assess statistically significant differences in HR, RR, and RT at T1, T2, and T3. The normality distribution and variance of data were evaluated by Shapiro–Wilk (swilk) and variance comparison test (sdtest) packages. The statistical significance threshold was set at p value <0.05. The statistical analysis was carried out by Stata Statistical Software Release 16 (StataCorp LLC, USA) [51].

## 3. Results

Twenty-one Apennine chamois, including eighteen females (one yearling, four subadults, and thirteen adults) and three males (one yearling and two adults) were captured. Eighteen chamois were physically immobilized in five electro-mechanical trap capture operations (3.6 chamois per operation, range 2–5), and the remaining three were captured in three box trapping events (one chamois per capture operation). 

Once entangled, the chamois were reached in 7.62 ± 2.22 (mean ± standard deviation) minutes (min 3.0 min; max 11.0 min) both in the case of the collective captures by means of electro-mechanical trap and in the individual captures in box-traps. All the captured chamois were apparently in good physical condition, weighing 26.2 ± 5.01 (mean ± standard deviation, min. 15.1 kg, max 38.0 kg), and no disease findings were observed at clinical examination. The mean anesthetic dosages of medetomidine, ketamine, and acepromazine actually received by the 21 Apennine chamois, obtained after weighing the animals in the field, are indicated in Table 2. 

The time of induction (T0) was 7.30 ± 2.16 (mean ± standard deviation) minutes (min 4.0 min; max 11.5 min). The average values and ranges for RR, HR, and TR at T1, T2, and T3 are reported in Table 3.

There were no statistically significant differences in RR, TR, or HR among the three measuring times (T1, T2, and T3, Table 3).

The time to awakening (T5) from the injection of the antagonist was 6.43 ± 3.04 (mean ± standard deviation) minutes (min 3.0 min; max 13.0 min) and the total period of anesthesia considered as the interval between T0 and T5 was 65.48 ± 38.78 (mean ± standard deviation) minutes (min 33.0 min; max 154.0 min). 

The values of the hematological and serum chemistry analyses are reported in Table 4, and the results of the venous blood gas analysis are described in Table 5. The different sample sizes (N) examined are due to the fact that the volume of blood taken was not always sufficient to evaluate all the parameters for each captured animal, given the context of multi-individual captures and logistic constraints on field sampling in mountain environments. 

No disease signs or lesions were detected during the entire study period. One chamois died during the capture operation, and another one, which had undergone anesthesia for 74 min, deceased 24 h after the release, accounting for a 4.8% immediate and a 4.8% delayed mortality rate, respectively. During necropsy, a C1-C2 spinal trauma was found in the chamois that died during the capture event, and lesions compatible with aspiration pneumonia were observed in the chamois recovered after capture.

## 4. Discussion

This study reports the first experience of collective physical capture followed by anesthesia and physiological monitoring in Apennine chamois, including blood gas analysis, in the field for the first time in the *Rupicapra* genus. The methods used in this study have been previously used to capture chamois [5,21,24,27,32], but the requirements of capturing a vulnerable subspecies for conservation purposes lead to modifications of the methodology and enhanced monitoring, namely: (1) the need for collective capture of females to achieve reproductive potential and philopatry and thus ensure the cohesion of the group [45,46]; (2) combining physical capture with anesthesia in order to smooth post-capture handling and decrease stress; (3) adding acepromazine to the anesthetic protocol as a tranquilizer and prevention for capture myopathy [22,39]; (4) clinical and physiological monitoring of the captured chamois, including, for the first time in the genus, venous gas analysis; and (5) monitoring the physiology and outcome of the chamois after release, both through hematological and serum biochemical analyses and with GPS/GSM radiocollars.

### 4.1. Performance of the Capture Method

The electro-mechanical trap allowed for the capture of chamois groups adequate for translocation and the formation of new colonies, with a higher performance (number of chamois captured per capture operation) than the box-trap (Table 1). This performance was also higher than that of the same method for Alpine chamois (*Rupicapra rupicapra rupicapra*) [24] and similar to the drive-nets used to capture Pyrenean chamois [23], agreeing with the reported higher selectivity and performance of operator-activated collective capture methods as compared to automatically-triggered methods [5,21,52]. 

### 4.2. Combination of Physical Capture and Anesthesia

Although physical trapping elicits a catecholamine stress response in chamois [19,22,53,54] that could interfere the effects of the anesthesia [55,56,57], the induction times overlapped with those reported for teleanesthesia without previous physical restraint in both Alpine and Apennine chamois, in which either xylazine–ketamine or romifidine–acepromazine–ketamine was used [32,58,59]. Anesthesia eased post-capture handling both for the chamois and the operator [5,21], and avoided the risk of cliff falls [60]. On the other hand, this protocol also added the risk of chemical immobilization to physical capture-related dangers. As a consequence, the mortality registered (2/21, 9.5%) could be attributed equally to the physical capture method and the anesthesia. Traumatic mortality related to the electro-mechanical trap has already been reported for physical capture of chamois [5,23,24], and this falls within the range of previously reported mortality for electro-mechanical trap- and net-captured roe deer (*Capreolus capreolus*) and chamois [5,21,23,24,61,62,63,64,65,66]. Chemical immobilization of free-ranging wildlife means anesthetizing in the worst possible conditions, since there is no preanesthetic evaluation of the animal and the on-field resources are limited. As a consequence, the previously reported mortality rates for anesthesia of wild free-ranging European mountain ungulates range from 0% to 26% [16,21,27,28,58,60,67,68]. However, the only anesthesia-related mortality event registered in our study was recorded through post-release monitoring, while in the previously reported cases, the mortality taxes were assessed only during the anesthesia and not after release (both for physical capture and anesthesia), and could, therefore, be underestimated as compared to our data. The post-release monitoring of the Apennine chamois captured and translocated in this study allowed us to confirm the long-term survival of 19 out of the 20 released chamois. Overall, the combination of physical capture and anesthesia captured and immobilized the chamois efficiently and safely, without further complications related to the use of either method.

### 4.3. Adding Acepromazine to the Medetomidine–Ketamine Anesthetic Protocol

Acepromazine has been used in captured mountain ungulates to decrease physiological stress and the probability of developing capture myopathy [22,39,41,43,69,70]. Acepromazine induces peripheric vasodilation through α-adrenergic blockade as a side effect [71], increasing tissue perfusion and preventing muscular and renal vasoconstriction, anaerobic metabolism, and hyperthermia, which are involved in the pathogenesis of capture myopathy [7,12]. Although the α-adrenergic effect of acepromazine could act synergically with the α2-agonist effects of medetomidine, and thus increase the risk of regurgitation of rumen contents and consequent aspiration pneumonia related to the relaxation of the cardia [72], the onset of acepromazine action takes at least 20 to 45 min after intramuscular injection and reaches its maximum after four to six hours [40,73,74,75]. Therefore, it is unlikely that acepromazine significantly enhanced the α2-agonists effects of medetomidine during the anesthesia (including the chamois deceased by aspiration pneumonia, which had been anesthetized for 74 min). Instead, acepromazine probably contributed to the ease of handling the chamois and improved animal welfare during the post-anesthetic containment and transport, which increases the physiological stress of the chamois [22] (the chamois were clinically stable and low-reactive, with reduced movements and absence of tachypnea, restlessness, or panic reactions while standing in the transport box during the noisy helicopter flight).

### 4.4. Venous Gas Analysis

To the authors’ knowledge, this is the first time that blood gas analysis has been reported for the clinical monitoring of free-ranging chamois or any other mountain ungulate species in Europe. Although field conditions limited the sampling to venous blood instead of arterial blood, the values of oxygen saturation in venous blood (SvO_2_) can be considered more suitable indicators than the previously available data obtained for Apennine chamois through pulse oximetry alone [59], which is not reliable in chamois because of their pigmented tongue. Furthermore, the supply of medical oxygen from portable tanks to the chamois in this study for at least 10 minutes between 15 and 45 min of anesthesia could further explain the higher oxygen saturation values which were obtained. The remaining results of the blood gas analysis agreed overall with the posterior laboratory analyses (discussed below) in detecting moderate metabolic acidosis, severe hyperlactatemia leading to base excess, and a decrease in circulating bicarbonate ions (HCO_3_^−^) (Table 5). This is compatible with capture stress [22,41,70], but blood gas analysis allows for immediate real-time monitoring and knowledge of the physiological and metabolic statuses of the captured individuals. On-field gas analysis should be further standardized to establish thresholds for handling and treatment decisions, such as fluid therapy, correction of acid–base imbalances, hypothermic therapy, and/or release of the captured animals, which should improve animal welfare and survival when capturing wild mountain ungulates.

### 4.5. Physiological Monitoring

The hematological and serum biochemical analyses identified the physiological effects of the sympathetic adrenergic response to physical capture buffered by the posterior anesthesia, as compared to the previously reported reference values for Pyrenean chamois [19]. Thus, red blood cell counts (5.53 × 10^12^/L vs. 14.69 × 10^12^/L) and hematocrit (0.277 L/L vs. 0.48 L/L) were lower than the previously reported values, probably because anesthesia prevented the stress-related release of catecholamines after physical capture and the consequent adrenergic contraction of the smooth muscle of the splenic capsule [22]. However, the predominance of lymphocytes was consistent with the adrenergic phase of stress response. 

The biochemical analyses indicated muscular damage induced by capture stress, since the muscular enzyme activities (CK 968.59 UI/L vs. 1093.1 UI/L; AST 992.25 UI/L vs. 274.2 UI/L; ALT 57.62 UI/L vs. 27.2 UI/L; ALP 576.62 UI/L vs. 465.5 UI/L), potassium (6.67 mmol/L vs. 5.56 mmol/L), and lactate (18.00 mmol/L vs. 20.75 mmol/L) concentrations were similar or higher in the physically captured and anesthetized Apennine chamois than in the physically captured Pyrenean chamois [18,19] or the teleanesthetized Apennine chamois [59]. Lactate is the result of anaerobic metabolism, and serum values of muscular enzyme activities and potassium concentration increase due to leakage caused by muscular cell membrane permeability and rupture related to the hypoperfusion, metabolic acidosis, decrease in heat dissipation, and hypoxia induced by catecholamine vasoconstriction as part of the pathogenesis of capture myopathy. Moreover, hyperpotassemia sensitizes the myocardium to the action of catecholamines, being responsible for the delayed peracute syndrome, the fourth syndrome of capture myopathy [7,12,41,43]. On the other hand, the indicators of renal perfusion, namely urea (6.48 mmol/L vs. 9.58 mmol/L) and creatinine (72.76 µmol/L vs. 113.01 µmol/L), were lower in the physically captured and anesthetized Apennine chamois than in the physically captured Pyrenean chamois [19], indicating that anesthesia prevented catecholamine-induced vasoconstriction in the renal arterioles more efficiently than in the muscle [41,43,68,70]. Moreover, serum muscular enzyme activities, as well as potassium, lactate, urea, and creatinine concentrations, continued to increase in Pyrenean chamois when restraint and immobilization time increased [22], further suggesting that anesthetizing the Apennine chamois with a combination including acepromazine was useful for disrupting the metabolic compromise and increasing tissue perfusion during the necessarily long handling, translocation, and release procedures. Therefore, it was shown that anesthesia enhanced animal welfare and increased the possibility that the chamois would survive and successfully adapt to the new environment. Finally, other serum biochemical variables, such as serum glucose, triglyceride, cholesterol, total protein, and sodium concentrations, were overall similar to the values previously reported for Pyrenean chamois [19].

## 5. Conclusions

The combination of physical capture with anesthesia reduced the physiological stress related to capture that would typically lead to capture myopathy in Apennine chamois, as indicated by hematological variables and serum biochemical analytes related to renal catecholamine-induced vasoconstriction. No negative side effects of adding acepromazine to the anesthetic protocol were detected, and it probably further decreased physiological stress and the probability of developing capture myopathy, as previously reported in physically captured mountain ungulates [22,39,41,50,68,69,70]. The use of field blood gas analysis for the first time on mountain ungulates in Europe allowed for real-time monitoring of the physiological status of captured wildlife. Post-release GPS monitoring allowed for the detection of mortality, which would otherwise have been unnoticed, and made it possible to ensure that the remaining chamois survived and adapted to their new environment after translocation. This mixed capture method, combined with the most complete clinical monitoring and emergency prevention reported to date in capture operations of European mountain ungulates, can be effectively used in conservation and management programs requiring collective captures and/or translocations.

## Figures and Tables

**Table 1 animals-13-00460-t001:** List of captured Apennine chamois.

N.	Sex	Capture Event	Physical Immobilization
1	F	09/08/2012	upnet
2	F	09/08/2012	upnet
3	M	09/08/2012	upnet
4	F	09/08/2012	upnet
5	F	29/08/2012	modified box trap (cave)
6	F	17/07/2013	upnet
7	F	17/07/2013	upnet
8	F	17/07/2013	upnet
9	F	18/07/2013	modified box trap (cave)
10	F	24/07/2013	upnet
11	F	24/07/2013	upnet
12	F	24/07/2013	upnet
13	F	24/07/2013	upnet
14	F	24/07/2013	upnet
15	F	29/07/2013	box trap
16	F	23/08/2013	upnet
17	F	23/08/2013	upnet
18	M	08/08/2014	upnet
19	F	08/08/2014	upnet
20	M	08/08/2014	upnet
21	F	08/08/2014	upnet

**Table 2 animals-13-00460-t002:** Mean anesthetic dosages administered to the captured Apennine chamois.

N	Dose (Mean ± Standard Deviation)		
	Medetomidine (mg/kg)	Ketamine (mg/kg)	Acepromazine (mg/kg)
21	0.054 ± 0.007	2.14 ± 0.28	0.043 ± 0.006

**Table 3 animals-13-00460-t003:** Respiratory rate, heart rate, and rectal temperature values of the captured Apennine chamois.

Value	Time	N	Mean + SD	Range
RR (rpm)	T1	16	74.9 ± 21.4	33–101
	T2	11	68.4 ± 20.2	37–97
	T3	7	65.7 ± 20.1	36–91
HR (bpm)	T1	14	111.5 ± 28.4	85–178
	T2	11	109.0 ± 52.7	48–218
	T3	7	116.4 ± 66.7	53–243
TR (°C)	T1	20	40.9 ± 1.0	39.2–43.0
	T2	12	40.7 ± 1.3	38.1–42.5
	T3	6	41.3 ± 1.1	40.0–42.8

N: sample size; SD: standard deviation; RR: respiratory rate; HR: heart rate; TR: rectal temperature; rpm: respirations per minute; bpm: beats per minute.

**Table 4 animals-13-00460-t004:** Hematological and serum chemistry data of the captured Apennine chamois.

Sample	Variable (Unit)	N	Mean ± SD	Median	Central 95% Interval	Range
Blood	RBC (×10 ^12^/L)	15	5.53 ± 0.55	5.62	4.30–6.26	4.12–6.26
	HGB (g/L)	15	177.3 ± 16.6	177	154.5–211.0	151–211
	HCT (L/L)	15	0.277 ± 0.030	0.281	0.212–0.319	0.203–0.319
	MCV (fL)	15	50.14 ± 0.48	50.1	49.37–51.00	49.3–51.0
	MCH (pg)	15	32.41 ± 3.89	31.9	28.19–41.03	27.8–43.2
	MCHC (g/L)	15	646.9 ± 81.3	634	563.05–832.25	555–876
	MPV (fL)	15	9.75 ± 2.74			7.1–13.6
	WBC (×10^9^/L)	15	12.30 ± 4.87	10.8	4.93–18.98	4.4–19.4
	LYM (×10^9^/L)	15	6.46 ± 2.33	6.3	2.91–10.80	2.7–10.8
	MID (×10^9^/L)	15	1.21 ± 0.46	1.2	0.54–2.03	0.5–2.2
	GRAN (×10^9^/L)	15	4.63 ± 3.09	4	1.17–10.07	1.1–10.1
	LYM (%)	15	54.90 ± 12.46	57.1	34.06–72.20	32.8–72.2
	MID (%)	15	10.09 ± 2.55	9.6	6.58–14.37	6.4–14.4
	GRAN (%)	15	35.01 ± 12.81	31.3	18.66–57.89	17.5–60.3
Serum	Glucose (mmol/L)	13	6.83 ± 2.73	6.72	3.83–10.64	3.83–10.94
	Cholesterol (mmol/L)	4	2.98 ± 1.02	2.83	2.04–4.18	2.01–4.24
	Triglyceride (mmol/L)	4	1.29 ± 0.32	1.28	0.97–1.63	0.96–1.64
	Urea (mmol/L)	13	6.48 ± 1.04	6.06	5.21–8.46	5.00–8.57
	Creatinine (µmol/L)	13	72.76 ± 21.40	70.7	40.66–112.27	35.36–114.92
	Total bilirubin (µmol/L)	13	4.08 ± 1.64	5.13	0.51–5.13	0.00–5.13
	Total protein (g/L)	13	69.7 ± 5.7	69	59.4–78.1	57–79
	Albumin (g/L)	13	31.4 ± 2.5	31	27.3– 35.0	27–35
	Globulins (g/L)	13	39.8 ± 6.3	41	31.3– 47.0	31–47
	CK (UI/L) *	9	968.59 ± 393.26			
	ALT (UI/L)	13	57.62 ± 21.22	49	38.0– 99.3	38–108
	AST (UI/L)	4	992.25 ± 710.95	1030	159.48–1760.85	103–1806
	ALP (UI/L)	13	576.62 ± 157.41	567	292.7–825.4	245– 865
	GGT (UI/L)	4	257.38 ± 167.97	245	77.83– 457.96	66.3– 473.2
	Amylase (UI/L)	13	29.92 ± 11.94	28	13.9–53.4	13–57
	Calcium (mmol/L)	13	2.77 ± 0.33	2.74	2.33–3.27	2.30–3.27
	Phosphorus (mmol/L)	11	1.91 ± 0.64	2.06	0.91–2.69	0.84–2.71
	Sodium (mmol/L)	11	139.55 ± 3.33	139	134.50–144.75	134–145
	Potassium (mmol/L)	6	6.67 ± 0.91	6.25	5.84–7.95	5.8–8.0

N: sample size; SD: standard deviation; RBC: red blood cell count; HGB: hemoglobin concentration; HCT: hematocrit; MCV: mean corpuscular volume; MCH: mean corpuscular hemoglobin; MCHC: mean corpuscular hemoglobin concentration; MPV: mean corpuscular volume; WBC: white blood cell count; LYM: lymphocytes; MID: monocytes; GRAN: granulocytes; CK: creatin kinase (* the CK activity was determined in nine chamois by the University of Milan within the Life Coornata Project [3,41]); ALT: alanine aminotransferase; AST: aspartate aminotransferase; ALP: alkaline phosphatase; GGT: gamma-glutamyltransferase.

**Table 5 animals-13-00460-t005:** Results of venous blood gas analysis of the captured Apennine chamois (*n* = 7).

Variable	Mean ± SD	Range
pH	6.99 ± 1.75	6.73–7.22
Bicarbonate ion (HCO_3_^−^, mmol/L)	9.31 ± 3.52	2.9–14.7
Lactate (mmol/L)	18.00 ± 9.05	15.82–19.75
Base excess (mmol/L)	– 22.13 ± 7.09	– 30–13
Anion gap mmol/L	26.33 ± 10.48	21–30
PvO_2_ (mmHg)	116.67 ± 42.73	61–163
PvCO_2_ (mmHg)	37.52 ± 11.23	21.7–47.9
SvO_2_ (%)	89.78 ± 10.52	70–90
Hematocrit (L/L)	0.403 ± 0.032	0.34–0. 44
Hemoglobin concentration (g/L)	137.1 ± 10.0	116–150
Glucose (mmol/L)	6.29 ± 0.88	5.0–7.6
Urea (mmol/L)	8.35 ± 1.32	7.13–11.05
Sodium (mmol/L)	143.14 ± 5.81	136–151
Potassium (mmol/L)	5.63 ± 1.29	4.1–7.7
Chloride (mmol/L)	112.86 ± 6.23	107–123

SD: standard deviation; PvO_2_: mixed venous oxygen tension; PvCO_2_: mixed venous carbon dioxide tension; SvO_2_: mixed venous oxygen saturation of hemoglobin.

## Data Availability

The data presented in this study are available on request from the corresponding author at Wildlife Research Center—Maiella National Park, Italy, simone.angelucci@parcomajella.it.

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
