# Peer review of "Welfare and Clinical Assessment on Physical Captures Followed by Anesthesia in Apennine Chamois (*Rupicapra pyrenaica ornata*)"

_animals, 2023, doi:10.3390/ani13030460_

Round 1

Reviewer 1 Report

Comments:

Apennine chamois (Rupicapra pyrenaica ornata) is a vulnerable but valuable subspecies. This study represents an innovative approach for collective mountain ungulate captures, useful to improve reintroduction activities for conservation purposes. The aim of this study was to evaluate the physiological effects on free-ranging Apennine chamois of collective physical capture followed by chemical im-mobilization with acepromazine-medetomidine-ketamine for reintroduction activities. Overall, this manuscript was well-written and in good shape. Questions or comments regarding this manuscript are as follows:

Line 98. The statement was not consistent with Line 181. How many Apennine chamois had been captured? How many were carried out with up-net?

Line 121. Please distinguish the time points and the time periods. HR, RR and Temperature should be recorded every 5 min, because the body conditions have changed. Recorded or analyzed the mean HR, RR, Temperature in time periods (T1 (15-25min), T2 (34-37min), T3 (45-54min)) were not accurate and did not make sense.

Line 145. Please add the time points of collecting blood samples for hematological and biochemical analyses. The period of anesthesia can influence the data, especially the results of blood gas.

Line 196. The time of induction was defined as the interval between injection time and the time that the animal become laterally recumbent and could be handled safely with minimal physical response, not monitoring (T0).

Line 199. Please list the exact time point of monitoring and reanalysis the data. I suggest to add a curve to show the indicators (HR, RR, Temperature) changed over time.

Line 206. The total period of anesthesia was from the time of injected anesthetics to the time of recovery rather than the interval between start to monitoring (T0) and awakening time (T5).

Line 205-208. Were all Apennine chamois injected the antagonist? What was the actual dosage for each individual? Were any additional anesthetics used for the one immobilized 154 min? Or did it recover from anesthesia without antagonist? The statistics of ‘the total period of anesthesia’ (mean ± SD and the range) does not make sense. We cannot determine the real duration of this anesthesia protocol and dosage.

Line 211. Why the N of Serum are different?

Line 239-387. There were no limitations in discussion. Please choose the main points to discuss and rewrite the discussion.

Line 395. I think this was not something you can conclude from this study: ‘…adding acepromazine to the anesthetic protocol probably improved chamois homeostasis during handling and transport and eased post-release adaption to the new environment and survival.’ In this study, you didn’t compare the effects of anesthesia protocol with or without acepromazine. Please rewrite the conclusions.

Author Response

Response to Reviewer 1 Comments

Apennine chamois (Rupicapra pyrenaica ornata) is a vulnerable but valuable subspecies. This study represents an innovative approach for collective mountain ungulate captures, useful to improve reintroduction activities for conservation purposes. The aim of this study was to evaluate the physiological effects on free-ranging Apennine chamois of collective physical capture followed by chemical im-mobilization with acepromazine-medetomidine-ketamine for reintroduction activities. Overall, this manuscript was well-written and in good shape. Questions or comments regarding this manuscript are as follows:

Line 98. The statement was not consistent with Line 181. How many Apennine chamois had been captured? How many were carried out with up-net?

Twenty-one Apennine chamois were captured in eight capture sessions, using physical capture methods followed by chemical immobilization. Five capture operations were carried out with up-net®, with 18 chamois captured, and three capture operations with box traps, with three were captured in box traps overall. To avoid potential confusion by readers, the number of capture operations carried out with each capture method has been repeated now in lines 195-199.

Line 121. Please distinguish the time points and the time periods. HR, RR and Temperature should be recorded every 5 min, because the body conditions have changed. Recorded or analyzed the mean HR, RR, Temperature in time periods (T1 (15-25min), T2 (34-37min), T3 (45-54min)) were not accurate and did not make sense.

Thank you for pointing out this aspect. Field conditions, including the simultaneous capture of more than one chamois, prevented consistently recording clinical values at exactly the same time throughout all the capture events. However, the aforementioned clinical variables (HR, RR and Temperature) were recorded for all the capture chamois at the same time intervals (at T1, between 15 and 25 minutes after T0; at T2, between 4-37 minutes after T0; and at T3, between 45 and 54 minutes after T0).Since the time frame was so narrow for each interval and consistently apart from the other intervals, the authors believe that data recording was representative enough to detect any trend in these variables.

 Line 145. Please add the time points of collecting blood samples for hematological and biochemical analyses. The period of anesthesia can influence the data, especially the results of blood gas.

 As reported in line 140, blood samples were collected from the jugular vein between 15 and 45 minutes after T0. The intervals between blood collection and analysis have now been provided for hematological analyses (line160) and blood gas analyses (line 178). Admittedly, both hematological and blood gas analysis values can vary within this 30-minute time frame. Nevertheless, taking into account that this is the first time blood gas analyses values are reported for a free-ranging mountain ungulate in Europe and the difficult operating conditions in mountain environments, the authors believe this information is worth publication and that the variation due to the wide time frame of sampling can be assumed to report this innovation. Maybe future research could be able to clarify whether the variation in these values due to this time frame can be higher than expected or not.

Line 196. The time of induction was defined as the interval between injection time and the time that the animal become laterally recumbent and could be handled safely with minimal physical response, not monitoring (T0).

Induction time is defined in lines 127-129 “as the time interval between the hand administration of the anesthetic mixture and the absence of voluntary movements”, and the sentence (in line 196 of the previous version) mentioned by the reviewer (nor other parts of the manuscript) does not contradict it. In fact, the clinical variables (HR, RR, and Temperature) were recorded at specific time intervals after T0 (absence of voluntary movements and not monitoring), as follows: T1 (15-25 min), T2 (34-37 min), and T3 (45-54 min). 

Line 199. Please list the exact time point of monitoring and reanalysis the data. I suggest to add a curve to show the indicators (HR, RR, Temperature) changed over time.

As aforementioned, it is not possible to list the exact time point of monitoring physical variables due to the field conditions during capture and clinical data recording. The authors acknowledge that plotting the variables in a curve would produce a nice graphical result. However, the dispersion of the values recorded suggest that such an esthetic output would not correspond with relevant statistical findings, since interindividual variability would overcome the effect of time in the values for these three clinical indicators.

 Line 206. The total period of anesthesia was from the time of injected anesthetics to the time of recovery rather than the interval between start to monitoring (T0) and awakening time (T5).

 Anesthesia is defined as insensitivity to pain (Oxford Language Dictionary), a state of controlled unconsciousness during which you feel nothing (Royal College of Anaesthetists) or similarly by other language, academic or medical institutions. Consequently, injecting, orally administering, or providing through gas a drug does not mean that the individual receiving it is “anesthetized”, but it will only be so once the drug produces the desired effect. Consequently, in this paper the total period of anesthesia was not evaluated starting from the time of injection but from the induction time (T0): as noted above, the induction time was recorded as the time interval between the administration of the anesthetic mixture and the absence of voluntary movements of the chamois. The time of induction (T0) was 7.30 ± 2.16 (mean ± standard deviation) minutes (min 4.0 min - max 11.5 min). We consider the total period of anesthesia as the interval between T0 (the recognition of absence of voluntary movements of the chamois, that is the end of the interval of the induction time) and T5: this period (the period from the moment the animal falls asleep to the moment the animal awakens) was 65.48 ± 38.78 (mean ± standard deviation) minutes (min 33.0 min - max 154.0 min).

Line 205-208. Were all Apennine chamois injected the antagonist? What was the actual dosage for each individual? Were any additional anesthetics used for the one immobilized 154 min? Or did it recover from anesthesia without antagonist? The statistics of ‘the total period of anesthesia’ (mean ± SD and the range) does not make sense. We cannot determine the real duration of this anesthesia protocol and dosage.

 All the chamois were injected with atipamezole as antagonist. Atipamezole (Antisedan®, Vetoquinol, Italy) at 0.12-0.16 mg/kg was intramuscularly injected to reverse sedation at time T4.

The chamois showing the highest value of anesthesia time (154 min.) was not supplemented but only antagonized very late. This chamois was captured in a group using up-net® where all the chamois were antagonized later than usual, due to logistical constraints related to the arrival of the helicopter. This event refers to an exceptional case, because in other events of delayed helicopter arrival the antidote was administered in normal times and the chamois waited awake in the wooden boxes for the helicopter. Please find below the time of antidote administration and the awakening time of this chamois.

Chamois name

Capture day

Method

Induction time T0

Antidote admin. T4

Awakening time T5

Time of anesthesia

Nicole

09/08/2012

upnet

17

166

171

154

Nevertheless, the authors do not agree with the statement by the reviewer that “The statistics of ‘the total period of anesthesia’ (mean ± SD and the range) does not make sense”. When using anesthetic combinations that can be totally or partially reversed, the duration of the anesthesia depends on the time the antagonist is administered. However, if no antagonist is administered, the effect of the anesthetic drug eventually fades-off and the animal awakes. Keeping wild animals anesthetized in the wild just to assess the duration of the anesthetic effect of a drug combination is ethically unacceptable, particularly in a threatened species like the Apennine chamois, since longer anesthesia times mean longer risks of anesthesia-related complications such as hypotension, bloating, respiratory depression, etcetera. However, this logistic incident allowed to compile the information that, in absence of the administration of an antagonist, the effects of the anesthetic combination used could last for two and a half hours. Whether keeping a chamois anesthetized for so long is advisable or not belongs to other type of studies and considerations regarding animal welfare, logistic constraints, professional ethic thresholds, etc., but undoubtedly this information is valuable and makes sense in the context of this study, in the opinion of the authors.

Line 211. Why the N of Serum are different?

The logicstic constraints of on-field sampling in mountain environments prevented the obtention of enough volume of serum to determine all the variables analysed in all the captured chamois.

Line 239-387. There were no limitations in discussion. Please choose the main points to discuss and rewrite the discussion.

Thank you for your suggestion, the discussion has been revised in the final text.

The Discussion section has been critically revised and shortened from 1761 words to 1624 words, even if a new paragraph on the pathogens investigated was added at the end of the Discussion section following the indications by reviewer #2". 

 Line 395. I think this was not something you can conclude from this study: ‘…adding acepromazine to the anesthetic protocol probably improved chamois homeostasis during handling and transport and eased post-release adaption to the new environment and survival.’ In this study, you didn’t compare the effects of anesthesia protocol with or without acepromazine. Please rewrite the conclusions.

Thank you for your suggestion, the conclusions have been revised in the final text.

The authors agree with the reviewer that the comparison between two anesthetic protocols, one with acepromazine and another without acepromazine, was not carried out in this study. However, the effects of acepromazine to reduce physiological stress and the probability to develop capture myopathy have been repeatedly reported in mountain ungulates, including the same species (Rupicapra pyrenaica). Since adding acepromazine to the anesthetic protocol can be useful to improve the health and welfare of captured chamois even if anesthetized, the authors believe that including this statement in the conclusions section, even if supported by references, is worth both for the scientific and the conservation management community.

Reviewer 2 Report

Reviewer comments on Manuscript number: animals-2085214

The present manuscript showed the effect of acepromazine-medetomidine-ketamine drug combination as chemical immobilization method on some physiologic and hematological values in Apennine chamois (Rupicapra pyrenaica ornata)

Broad comments

The data showed is potentially useful, as there is a lack of current literature about the use of acepromazine-medetomidine-ketamine drug combination as part of injectable anesthesia protocols for immobilizing Apennine chamois; however, several flaws were observed in the study design that seriously affected the results and conclusions of the manuscript.

Weakness of the study: The aim of the study is not clear.

Specific comments

Abstract

The abstract must include a summarize of the results of the study, that is missing in this section.

Keywords. Please remove up-net, and replace anesthesia for chemical restraint

Introduction.

The introduction does not support the proposed aim of the study. There is no background information about chemical restraint, nor sedatives – anesthetic agents used in Rupicabras. It is not clear if the main aim of the study is related to chemical immobilization, capture (mechanical nets) or biochemical blood values.

Material and methods.

L101. Once Up-net trap has been described, then electro-mechanical trap should be then used throughout the document. It seems that Up-net trademark has sponsored the study.

L158-161. Please the time elapsed between blood sampling and processing at laboratory (not the i-stat blood gases samples). Please consider this time for further discussion.

L177-179. Statistics. Considering the low number of animals per group and the wide range and limits of your data, I wonder if your data was non-parametric. Did you perform a normality test? Bartlett for variance equality ? A repeated measures analyses to observe differences between time points? Statistical package ? All these important information is missing in this section.

 Results.

Table 4. To this reviewer, there are important missing information in this table. If you are reporting range, then the median has to be also reported. I will make totally sense if your data is non-parametric.

No word said about the N numbers different to the initial 19 animals in table 4. Why only 7 samples for blood gases in table 5?

L221 – 231. Serological results if reported here should be fully discussed in the discussion section, otherwise should be removed from the study.

Discussion.

The discussion section has to be re-written, considering the above statements and suggested changes. Literature cited at this point should be related at least to other kind of injectable protocols in Rupicabras, this topic needs further discussion.

L331. Pulse oximetry can be assessed on mucous membranes.

L353-360. Medetomidine does not produce vasodilation as an alfa-2 agonist.

Conclusions.

They have to be re-written and shortened considering the aim of the study.

Thanks

Reviewer 3 Report

This is a well written and useful contribution to the literature on wildlife chemical immobilization and anesthesia. I have only minor comments:

Lines 53-55: No need to cite 9 references - some of them are either obsolete or irrelevant; delete refs 4-8 and add the most recent review: https://doi.org/10.1007/s11259-022-10030-9

Lines 99-100: As far as I can see, the cited ref (32) is about darting only and not the use of up-net

Reference list: This is the weak part of the manuscript.

All refs are not correctly formatted, e.g. author names in ref 40 and use of caps in titles of refs 41 and 42.

There are numerous outdated and/or irrelevant references. Delete refs 4-9, 13-15, 17, 19 and 22

Use the most recent edition of textbooks: Refs 39 and 59

Ref 32: There is a typo ('xilazine') and the published paper does not have page numbers (article number 567: https://doi.org/10.1007/s10344-009-0270-7

Ref 40: Typo (Scientific 'reports')

Author Response

Response to Reviewer 3 Comments

This is a well written and useful contribution to the literature on wildlife chemical immobilization and anesthesia. I have only minor comments:

Thank you for your positive overall assessment. We have tried to address all the comments by the reviewers in the best possible way.

Lines 53-55: No need to cite 9 references - some of them are either obsolete or irrelevant; delete refs 4-8 and add the most recent review: https://doi.org/10.1007/s11259-022-10030-9

The authors are grateful for this comment and have replaced the old references with more recent and adequate ones.

Lines 99-100: As far as I can see, the cited ref (32) is about darting only and not the use of up-net

Exactly, thanks to the suggestion, this reference has been corrected with [44], which concerns the up-net system.

Reference list: This is the weak part of the manuscript.

All refs are not correctly formatted, e.g. author names in ref 40 and use of caps in titles of refs 41 and 42.

There are numerous outdated and/or irrelevant references. Delete refs 4-9, 13-15, 17, 19 and 22

Use the most recent edition of textbooks: Refs 39 and 59.

Now [52] and [72].

Ref 32: There is a typo ('xilazine') and the published paper does not have page numbers (article number 567: https://doi.org/10.1007/s10344-009-0270-7

Ref 40: Typo (Scientific 'reports')

Thank you for the suggestion, the references have been corrected.

All references have been integrated, reordered, properly formatted and adapted to the most recent bibliography according to the advice of Reviewer#3.

Round 2

Reviewer 1 Report

I accept this version of manuscript.

Reviewer 2 Report

Reviewer comments on Manuscript number: animals-2085214 R2

The present manuscript showed the effect of acepromazine-medetomidine-ketamine drug combination as chemical immobilization method on some physiologic and hematological values in Apennine chamois (Rupicapra pyrenaica ornata)

Broad comments

The manuscript has been improved and just few things have to be addressed before publication.

Specific comments

Material and methods.

L177-179. Statistics. Please add trade mark, version and year of the statistical software used.

 Results.

Please specify in few lines the reason of different N numbers in tables 4 and 5.

L221 – 231. The aim of my previous comments about serological test results was related to the main objective of the study, that, as the authors have now stated, the serological results are not fully considered in that broad objective. Then, I would recommend to remove it from the manuscript. The authors may considerer to use this data for other publication.

Discussion.

L722-735. I would suggest, considering above comments, to remove the serological investigation section of the discussion.

Conclusions.

The conclusion has been improved. According to my previous comments, the serological data that are not showed in the conclusions, are aligned to the actual aim of the study, and that is correct.

Thanks

Author Response

Thank you for your attention.

Round 3

Reviewer 2 Report

Dear authors:

The manuscript has been improved according to this reviewer comments.

Good Luck